# Protection of the Liuzheng Water Source: A Karst Water System in Dawu, Zibo, China

**Henghua Zhu [1,2], Yanan Dong [3], Liting Xing [3,*], Xiaoxun Lan [3], Lizhi Yang [2], Zhizheng Liu [2] and Nongfang Bian [4]**

[1] School of Environmental Studies, China University of Geosciences, Wuhan 430074, China; zhuhenghua12@163.com

[2] Shandong Institute of Geological Survey, Jinan 250000, China; yanglizhi23@163.com (L.Y.); liuzhizheng12@163.com (Z.L.)

[3] School of Water Conservancy and Environment, University of Jinan, Jinan 250022, China; dongyanan96@163.com (Y.D.); jndxlanxx@163.com (X.L.)

[4] Administrative Department of Dawu Water Source Area of Zibo, Zibo 255400, China; biannongfang@163.com

* Correspondence: stu_xinglt@ujn.edu.cn; Tel.: +86-531-8276-9233

**Abstract:** The Dawu water source is a rare, large-scale groundwater source located in northern China. The water supply function from this water source has, however, been lost due to anthropogenic pollution. In order to fully utilize valuable groundwater resources, a new water source of urban domestic water in Liu Zheng is planned. In this study, a tracer test and a numerical simulation method are used to examine the hydraulic connection between the Liuzheng water source and the Wangzhai industrial park; to optimize the exploitation layout of the Liuzheng water source and Dawu water source; and to propose the extent of the Liuzheng water source protection area. Results indicate that: (1) Karst development in the study area is uneven, and the Wangzhai area is a recharge area of the Liuzheng water source; (2) it is predicted that the groundwater flow field will not be significantly changed when a groundwater volume of 150,000 $m^3$/day is exploited from the Liuzheng water source; (3) it is predicted that the proposed chemical park in Wangzhai will gradually pollute to the groundwater in the northern area of Liuzheng; and (4) results using the empirical formula method and the numerical simulation method indicate that the area of the primary protection area of the Liuzheng water source is about 0.59 $km^2$, and the area of the secondary protection area is about 14.98 $km^2$. Results from this study provide a certain technical basis for the exploitation and protection of groundwater in the Liuzheng water source.

**Keywords:** tracer test; numerical simulation; pollution; division of protection areas; Liuzheng water source

## 1. Introduction

The karst landforms are widely distributed around the world, and karst water provides drinking water to nearly a quarter of the world's population. In China, the total area of bare limestone is about 1.3 million square kilometers, accounting for 13.5% of the total area of the country, and covered limestone buried underground is more extensive. During the long geological history evolution, landforms such as gully, dissolved depression, dry valley, blind valley, falling water cave, and skylight are formed. The soluble rocks can be dissolved and eroded by groundwater along the layers, joints or structural fractures, and formed underground passages, rapidly moving through karst fissures, karst conduits, etc. Groundwater in the karst area is extremely easy to be polluted, which means that karst aquifer has a high degree of vulnerability [1–3]. The multiple conduit flow of karst medium leads to rapid contamination, while the diffuse flow forms persistent contamination [4]. Due to the

uneven development of karst as well as the complex and variable movement of groundwater flow, people try to study the migration of pollutants in karst water from different scales using different methods, such as isotopes [5], tracer test [6], pumping test [7], numerical simulation [8–10], indoor experiment [11], and so on. For example, Hamdan et al. [12] revealed the characteristics of groundwater vulnerability to pollution with the method of combining the stable isotopic (oxygen and hydrogen) and data of water temperature, spring discharge, and turbidity. Morales et al. [13] depicted the transport characteristics along preferential flow paths in karst aquifers by analyzing data obtained in 26 tracer tests in Basque.Maloszewski et al. [14] combined the application of lumped-parameter models to establish a measurement model for $^{18}O$ and tritium in precipitation and springs, and obtained mean values of hydraulic parameters. Therefore, in order to effectively manage vulnerable karst water systems, specific exploration techniques and modeling methods are required to master, studying the media characteristics and hydrodynamic field characteristics of the karst area [15,16].

Previous studies have indicated that the majority of water in karst areas with intense anthropogenic activity has been contaminated to different degrees [17–19]. Pollution sources are derived mainly from industrial, agricultural and domestic pollution [20], with industrial and agricultural pollution being the main source points. For example, industrial production activities produce refractory organic pollutants, heavy metals, etc., which cause different degrees of pollution to groundwater systems [21]. Currently, water pollution in karst areas has certain universality in the north and south of China, as well as in other karst areas around the world. Water pollution associated to the chemical industry in karst areas is a notably serious issue [22–24].

In order to protect groundwater resources and prevent groundwater pollution risks, methods suitable for the division of groundwater source protection areas have been examined [25–27]. For example, Andreo et al. [28] used karst water replenishment conditions combined with groundwater flow rate to delineate the risk zone for karst water pollution. With the rapid development of computer science, numerical methods have been widely used. Numerical methods can not only simulate large-scale karst water movementand evaluate karst water resources [1], but also predict groundwater pollution processes [29]. For instance, Rudakov et al. [8] used two-dimensional and three-dimensional flow to simulate the migration of pollutants in groundwater. The migration of characteristic pollutants in aquifers was investigated by Saghravani using FEFLOW [9], and Rao Lei et al. [10] used the Visual Modflow to simulate and study the migration of pollutants in groundwater after the leakage of sewage from an industrial park in Jiangjin District, Chongqing. In order to delineate groundwater protection zones, a large number of studies have been undertaken [30–33], with the majority using empirical methods to determine the extent of protected areas [34]. In order to predict the spread of pollutants over long-term scales [35], the numerical method has advantages over traditional empirical formula methods, however the numerical method requires more hydrogeological exploration work.

The Dawu water source is a rare large-scale karst water source. However, an oil spill by the Qilu Petrochemical Company in the mid-1980s resulted in groundwater pollution in this water source [36]. In order to control karst water pollution in this area, hydraulic interception measures were implemented in the upper reaches of the Dawu water source in 1992. However, these measures have not protected the Dawu water source as petroleum pollutants in some areas still exceed standard guideline values. At present, the Dawu water source has lost its water supply function. In order to alleviate water supply issues in this area, new water sources are urgently needed. According to previous analysis, the Dawu water source area and the upstream Liuzheng area belong to the same hydrogeological unit; the Liuzheng section in the runoff area has a large output capacity from a single well and it has good water quality [37]. It is therefore proposed that a new water source is created in the Liuzhen area. As a new petrochemical park is proposed in the Wangzhai area, it is important to study the impact this park may have on the groundwater environment of the Liuzheng water source.

Therefore, this paper uses the tracer test to reveal the characteristics of medium field, uses numerical simulation to grasp the characteristics of groundwater flow field, and combines the tracer test, empirical formula and numerical simulation to divide the protection area of karst water source area.

## 2. Study Area

The study area belongs toa warm temperate continental monsoon climate. The annual average temperature is 12.3–13.1 °C, and the average annual precipitation is 640.50 mm. The rivers in the area are all rain-source type, and the terrain is high in the south and low in the north. The terrain in the south is a low mountain and hilly, and the highest elevation is 380 m. The northern part is connected to the sloped plain in front ofa mountain, and the lowest elevation is 40 m. Cambrian carbonate rocks are exposed in the southern high mountains, central Ordovician limestones are exposed in the low hills and Ordovician limestones are concealed to the north in the Carboniferous-Permian sand sheets. Below the rock strata, Quaternary loose rock stratais widely distributed in the piedmont plain in the north, and the Carboniferous-Permian coal-bearing strata constitutes the water-blocking barrier of fractured karst water in the north. The fault structure is developed in the area. The central Zihe fault zone is the largest concealed fault zone in this area, which constitutes the water-conducting zone from the south to the north. The Liuzheng water source is located in the middle of this fault zone (Figure 1). The carbonate fractured karst aquifers in the southern region and the Quaternary loose-rock-porous aquifers in the northern region are the main aquifers in the study area. The Quaternary loose-rock-porous aquifer is mainly distributed in the floodplain of the Zihe River and along its two sides, the plain area in the north and the alluvial fan of the Zihe River. The carbonate fractured karst aquifer is mainly distributed in bare mountainous areas to piedmont concealed zones.

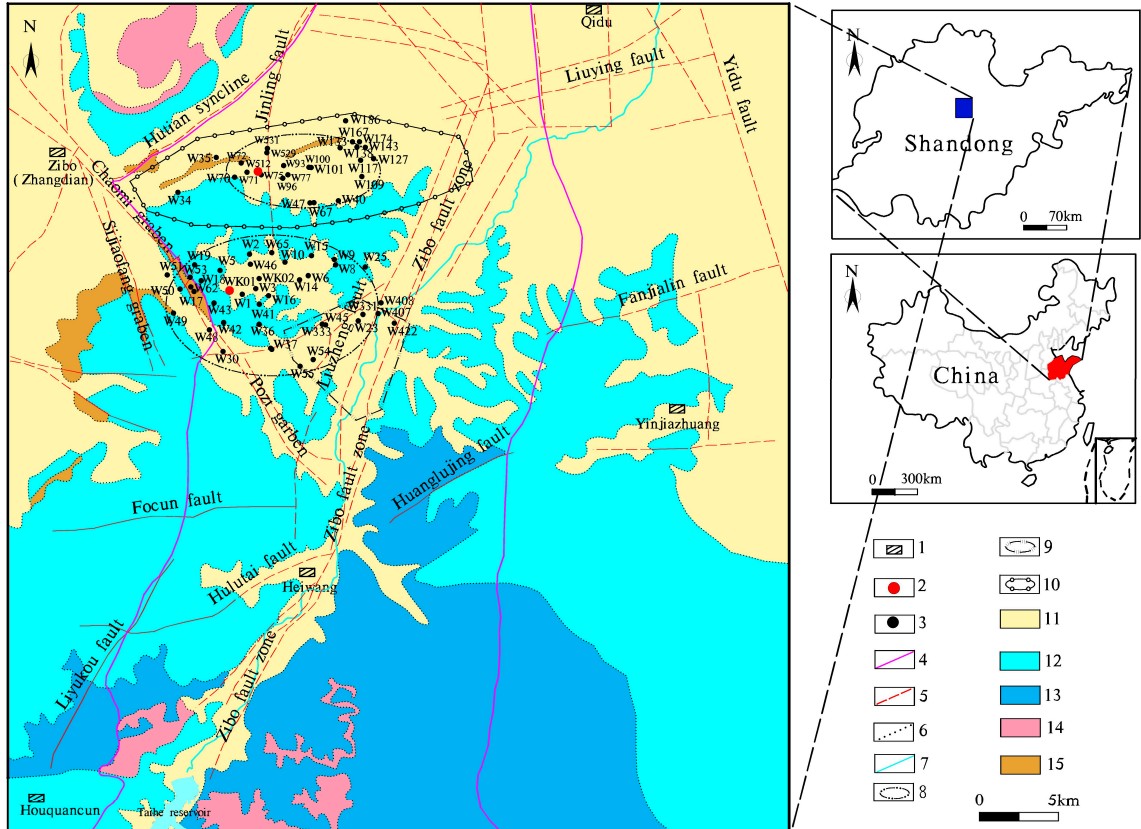

**Figure 1.** Geology of the study area. 1: Village; 2: Source well; 3: Observation well; 4: Simulation zone boundary; 5: Fault; 6: Geological boundary; 7: River; 8: Tracer test area; 9: Dawu water source exploitation area; 10: Liuzheng water source exploitation area; 11: Quaternary loose layer; 12: Ordovician limestone; 13: Cambrian limestone; 14: Magmatic rock; 15: Carboniferous-Permian sand shale.

Water recharge in the study area is mainly by atmospheric precipitation and water extraction is mainly due to artificial exploitation. Total water resources in the region are about 410,000 m³/day,

of which artificial groundwater exploitation is mainly concentrated in the northern region, with exploitation volume accounting for about 350,000 m$^3$/day [38]. However, groundwater pollution in this area caused by the oil spill in the 1980s has rendered it not suitable for domestic use. It is therefore necessary to optimize the exploitation layout of the Dawu and Liuzheng areas, and to divide the Liuzheng water source protection area with the aim to maximize the benefits of groundwater resources.

## 3. Methods

### 3.1. Tracer Test Arrangement

In order to examine the hydraulic connection between the Wangzhai, Hougao and Liuzheng areas, WK01 and W512 were used as source wells for field tracer tests (Figure 1). On 7March 2017, 210 kg of ammonium molybdate was injected into the WK01 well in the Wangzhai experimental area, and 27 monitoring wells were established. A total of 379 water samples were collected from the following day until 20June 2017. On 16November2017, 200 kg of ammonium molybdate was injected into the W512 well in the Hougao experimental area, and 25 monitoring wells were established. A total of 350 water samples were collected from the following day until 9March 2018. For both experiment sites, the aquifers of the source and sampling holes are Ordovician limestone aquifers.

Prior to the open tracer test, molybdenum ion background values from the Wangzhai and Hougao experimental areas were recorded.After the tracer was injected, in the encrypted observation area within 2 km from the source point, the monitoring frequency was once a day; the area outside the encryption area was sampled once every three days. Monitoring holes close to the source well were sampled first.As the tracer migrated, the sampling frequency of the surrounding area was gradually encrypted.The atomic absorption spectrophotometer-graphite furnace method was used to measure molybdenum ion concentrations, and the detection limit is 0.5 ppb. The water level measurement at the monitoring points was synchronized during the tracer test.

### 3.2. Groundwater Flow Model

According to the hydrogeological conditions of the study area, the aquifer in the study area was generalized into a heterogeneous, anisotropic, three-dimensional unsteady groundwater flow model. The mathematical model of groundwater flow can be written as [39]:

$$\begin{cases} \frac{\partial}{\partial x}\left(K_{xx}\frac{\partial H}{\partial x}\right) + \frac{\partial}{\partial y}\left(K_{yy}\frac{\partial H}{\partial y}\right) + \frac{\partial}{\partial z}\left(K_{zz}\frac{\partial H}{\partial z}\right) + W = \mu\frac{\partial H}{\partial t} \dots\dots\dots\dots (x,y,z) \in \Omega \\ H(x,y,z,t)|_{t=0} = H_0(x,y,z) \dots\dots\dots\dots\dots\dots\dots\dots\dots (x,y,z) \in \Omega \\ K_{xx}\frac{\partial H}{\partial x} + K_{yy}\frac{\partial H}{\partial y} + K_{zz}\frac{\partial H}{\partial z}|_{\Gamma_2} = q(x,y,z,t) \dots\dots\dots\dots (x,y,z) \in \Gamma_1 \end{cases} \quad (1)$$

where $K_{xx}$, $K_{yy}$ and $K_{zz}$ are values for hydraulic conductivity (m/day) along the $x$, $y$, and $z$ axes, respectively; $\mu$ is the specific storage or specific yield; $H$ (m) is the potentiometric head; $H_0$ (m) is the potentiometric head at the initial moment; $W$ is a volumetric flux per unit volume representing sources and/or sinks of water; $\Omega$ is the simulation area; and $\Gamma_1$ is the boundary.

According to the drilling data and hydrogeological conditions, the model was divided into three layers in the vertical direction: (i) The Quaternary pore aquifer; (ii) the relatively weak aquifer; and (iii) the fractured karst aquifer. The planar area was divided by a discrete method of rectangular finite difference and the cell size was 500 m × 500 m. The east-west boundary of the calculation zone was the water-blocking boundary; the north was the lateral runoff discharge; and in the south, the west side of the Taihe Reservoir was the lateral runoff recharge. The main recharge items in the study area were the recharge of precipitation infiltration, and other recharges include the lateral runoff recharge, river leakage recharge, and recharge of reservoirs during the flood season. The main extraction process is artificial exploitation. The initial flow field used the groundwater flow field on 1September 2018.

*3.3. Solute Transport Model*

In order to predict the pollution impact of the proposed chemical park on the groundwater of the Liuzheng water source, the mathematical model of solute transport is as follows [39]:

$$\begin{cases} \frac{\partial C}{\partial t} = \frac{\partial}{\partial x_i}(D_{ij}\frac{\partial C}{\partial x_i}) - \frac{\partial}{\partial x_i}(V_iC) + \frac{q_s}{n}C_s + \sum R_k \dots\dots\dots\dots\dots\dots (x,y,z) \in \Omega \\ C(x,y,z,t)|_{t=0} = C_0(x,y,z) \ \dots\dots\dots\dots\dots\dots\dots\dots\dots\dots (x,y,z) \in \Omega \end{cases} \quad (2)$$

where $C$ (mg/L) is the concentration of pollutants; $D_{ij}$ (m$^2$/day) is the hydrodynamic dispersion coefficient; $V_i$(m/day) is the groundwater permeation flow rate; $t$ (day) is time; $n$ is the porosity; $C_s$ (mg/L) is the concentration of source sink item; $q_s$ (m$^3$/day) is the unit flow; $\Sigma R_k$ is the chemical reaction term; $C_0$ (mg/L) is the initial concentration of pollutants; and $\Omega$ is the simulation area.

Based on the water flow model, the solute transport model was established. Combined with the composition of pollutants discharged from various chemical enterprises in the plant area [40], the conventional component Cl$^-$ was selected as the simulation predictor. Assuming the initial concentration of the region was 0, the pollution source was assumed to be a point source. Leakage was set to 5000 m$^3$/day, and the concentration of the pollutant Cl$^-$ was set to 3000 mg/L.

## 4. Results and Analysis

*4.1. Anisotropy of Karst Development*

### 4.1.1. Anisotropy of Runoff Channels

Tracer detection values were set at concentrations 5-times greater than background values at the observation wells. Molybdenum tracer was detected in eight observation wells in the Wangzhai test area and in ten observation wells in the Hougao experimental area. The tracer concentration curves form the observation wells in Wangzhai and Hougao were divided into single peak, double peak and triple peak types (Table 1), and the three major categories were subdivided into different specific forms. The single-peak curve was divided into slightly symmetrical and asymmetric types; the double-peak curve was divided into high-low and low-high types; and the three-peak curve was divided into high-low-low and low-high-low types.

Single peak concentration curves accounted for the majority of tracer curves (62.5%) from wells in the Wangzhai test area, followed by triple peak curve (25%) and double peak curve (12.5%). Results from observation wells in the Hougao test area recorded double peak (50%) and single peak (40%) as the dominant tracer concentration curves; the triple peak curve type accounted for 10%. The various types of curves reflect the difference in the degree of aquifer fissure development, with a single channel and two or three main channels [41]. The concentration curve obtained by the tracer test indicates that karst development in the Wangzhai and Hougao areas of Zibo are characterized by the coexistence of karst pores, karst fissures, fractures and pipelines. The range of tracer diffusion suggest that the main direction of the Hougao experimental area is northward and eastward, and the mainstream of the Wangzhai experimental area is to the southeast (Figure 2). Therefore, the Wangzhai section is a supply area of the Dawu and Liuzheng water sources.

**Table 1.** Tracer concentration curve type of observation well.

| Peak Type | Curve Type | Observation Well |
|:---:|:---:|:---:|
| Single peak | asymmetric type | Wangzhai: W43, W36<br>Hougao: W77 |
| Double peak | slightly symmetrical type | Wangzhai: W10, W45, W45<br>Hougao: W98, W96, W95 |
| | high-low type | Wangzhai: W8<br>Hougao: W76, W90, W101 |
| | low-high type | Hougao: W75, W529 |
| Triple peak | high-low-low type | Wangzhai: W16<br>Hougao: W531 |
| | low-high-low type | Wangzhai: W4 |

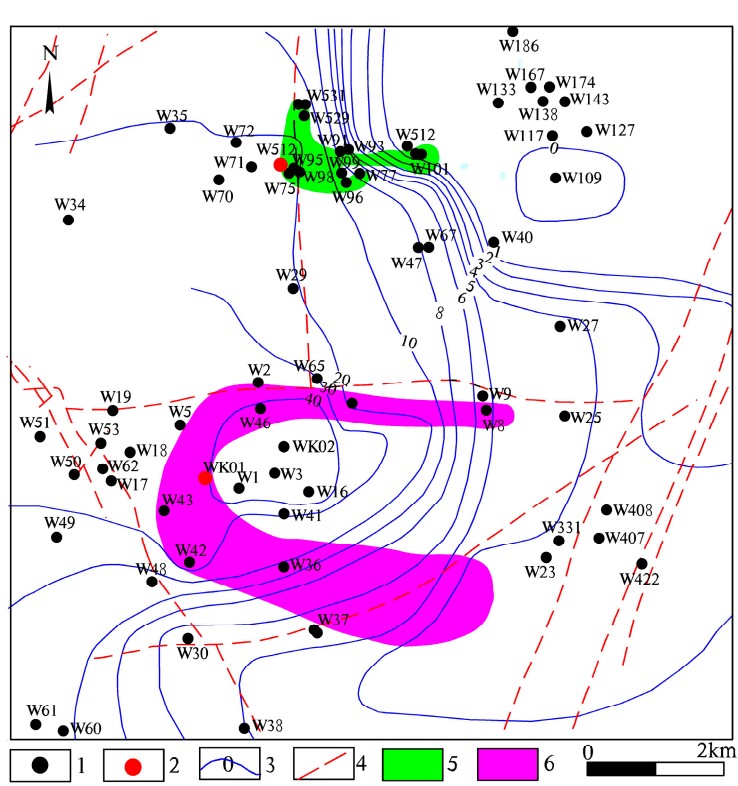

**Figure 2.** The range of tracer diffusion. 1: Observation well; 2: Source well; 3: Water level line/m; 4: Fault; 5: Tracer diffusion zone in Wangzhai; 6: Tracer diffusion zone in Hougao.

### 4.1.2. Anisotropy of Dispersion Coefficient

Assuming that the instantaneous injection of tracer in the steady flow field forms a two-dimensional dispersion, tracer migration will mainly be affected by mechanical dispersion, ignoring the influence of molecular diffusion [42]. The concentration distribution expression of a point (x, y) on the plane will therefore be:

$$c = \frac{M}{4n\pi\sqrt{\alpha_L \times \alpha_T}t} exp\left\{-\frac{1}{4ut}\left[\frac{(x-ut)^2}{\alpha_L} + \frac{y^2}{\alpha_T}\right] - \lambda t\right\} \tag{3}$$

where $c$ (mg/L) is the tracer concentration at $(x, y)$ point; $n$ is the porosity; $u$ (m/day) is the average pore flow velocity of the groundwater, and $\alpha_L$ (m) and $\alpha_T$ (m) are longitudinal and transverse dispersion, respectively.

According to the test data, it was possible to determine the peak moment of molybdenum ion concentration in the observation well ($t_m$), to organize the measured data with ($t_m^2/t+t$) as x and ($lnCt+t_m/t$) as y, and to the identify slope (B) and intercept (lnA) of the fitted line using linear regression. Similarly, the source hole data was collated and a scatter plot with $t$ as x and $lnCt$ as y was drawn. The intercept (ln$a$) of the fitted straight line was obtained. Then, these values were substituted into Formula 4, which can give the vertical and horizontal dispersion [43].

$$\begin{cases} \frac{x^2}{\alpha_L} - \frac{y^2}{\alpha_T} = 4u\left(Bt_m^2 + t_m\right) \\ \ln A - \ln a = \frac{x}{2\alpha_L} \\ B = \frac{u}{4\alpha_L} + \lambda \end{cases} \tag{4}$$

where $x$ (m) and $y$ (m) are the transverse distance and longitudinal distance from the observation hole to the source hole, respectively; $u$ (m/day) is the average pore flow velocity of the groundwater; and $t_m$ (day) is the time when the tracer concentration reached the peak time. The Equations are solved to obtain the longitudinal and transverse dispersion, which are substituted into the Equations $D_L = \alpha_L v$ and $D_T = \alpha_T v$ to obtain the longitudinal and transverse dispersion coefficients (Table 2).

**Table 2.** Calculation results of the dispersion coefficient.

| Source Well | Direction | Observation Well | Longitudinal Dispersion Coefficient (m²/day) | Transverse Dispersion Coefficient (m²/day) |
|---|---|---|---|---|
| | Southeast | W36 | 3.71 | 0.25 |
| WK01 | Southeast | W45 | 6.41 | 0.83 |
| (Wangzhai) | Southwest | W43 | 2.38 | 0.52 |
| | Northeast | W8 | 5.76 | 0.79 |
| | North | W31 | 12.83 | 0.93 |
| W512 | East | W101 | 3.06 | 0.4 |
| (Hougao) | East | W75 | 2.71 | 0.22 |
| | Southeast | W95 | 4.56 | 0.62 |

The dispersion coefficient in all directions of the test area is anisotropic, and it has no obvious correlation with the flow rate and tracer concentration. Therefore, the hydrodynamic field in this area is simulated using a three-dimensional flow model.

### 4.2. Water Level Prediction of the Liuzheng Water Source

According to the exploration data of the Liuzheng water source [38], the identification and inspection stage of the model was completed using data spanning June 2017 to September 2018. The permeability coefficient, elastic drainable porosity, specific yield and rainfall infiltration coefficient of each division of the study were also determined. Since the Liuzheng area is considered as a new water supply source, exploitation layout was appropriately adjusted according to previous regional groundwater exploitation volume. Therefore, the amount of exploitation in the northern

part of the Dawu area was reduced and the amount of artificial exploitation in the Liuzheng areawas increased. Precipitation data from 2008–2017, including the dry water series, was selected for our study. After debugging the exploitation capacity of the Dawu water and Liuzheng water sources several times, the dynamic change of the groundwater level in the Liuzheng area over the next 10 years was predicted based on an exploitation rate of 150,000 m$^3$/day.

Results indicate that when the total volume of the karst water system is kept constant, water extraction in the Liuzheng area is about 150,000 m$^3$/day, and 200,000 m$^3$/day in the Dawu water source. In this case, although the groundwater level of the Liuzheng is reduced in dry years, groundwater is quickly replenished and the water level is able to increase in wet years (Figure 3). The groundwater system can maintain a dynamic equilibrium for many years. Figure 4 showed the groundwater flow field of the Liuzhengarea during the wet season after adjusting the mining layout. There is no significant change in the regional natural groundwater flow field. The average water level in the mining center droppedfrom about 27m adjusting the mining layout to about 22mafter adjusting the mining layout, that is, the mean water level drop was about 5 m, and no hydrogeological problems will appear, such as karst collapse.

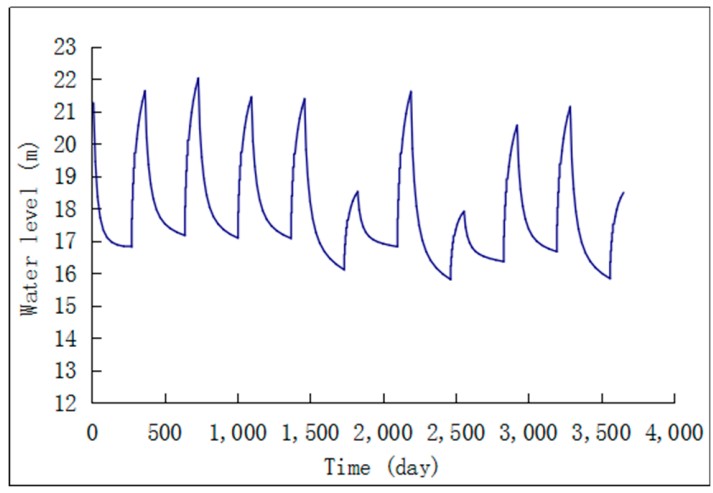

**Figure 3.** Prediction curve of water level change in the LK11 well.

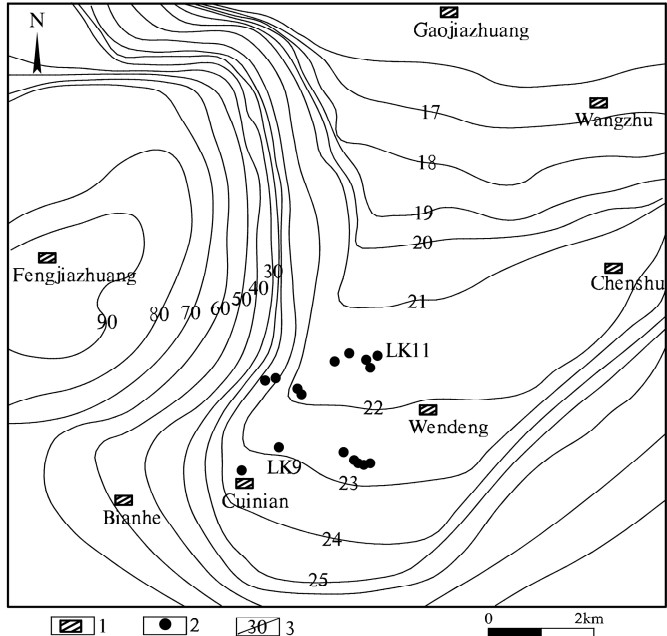

**Figure 4.** Groundwater plane flow field. 1: village; 2: mining well; 3: groundwater level contour (m).

After optimizing the exploitation layout, exploitation of high-quality groundwater in the Liuzheng area increased, and the source of inferior groundwater in the north of the Dawu water source was intercepted, thereby greatly improving the utilization efficiency of groundwater. These results indicate that the exploitation of 150,000 m³/day of water from the Liuzheng water source is feasible for domestic use.

### 4.3. Pollution Prediction

Based on the groundwater flow simulation when the amount of groundwater exploitation in the Liuzheng water source is 150,000 m³/day, the regional pollutant transport after 10, 30 and 50 years of continuous wastewater discharge (Figure 5), and the concentration curve of Cl⁻ in the mining well location of Liuzheng water source (Figure 6) were modeled.

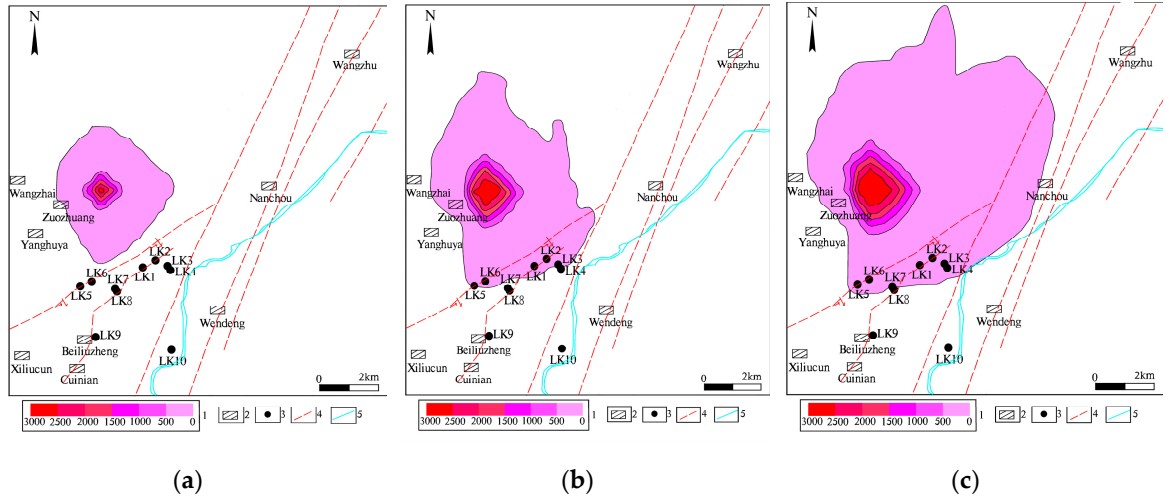

**Figure 5.** Plane range of solute migration. (**a**) 10 years, (**b**) 30 years, (**c**) 50 years; 1: Concentration of Cl⁻ (mg/L); 2: Village; 3: Liuzheng water source mining well; 4: Fault; 5: River.

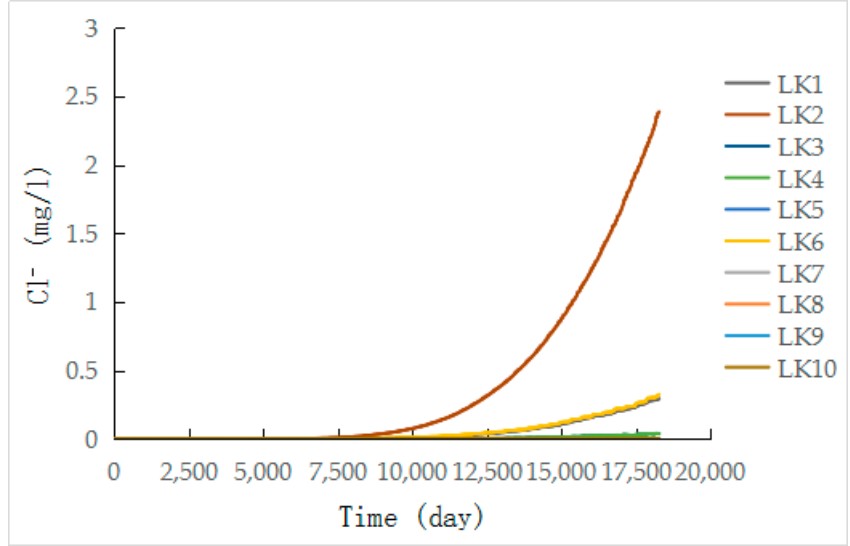

**Figure 6.** The concentration curve of Cl⁻.

Forecasts indicate that the solute will mainly migrate in a southeast direction, before gradually moving to the north in the main direction. After continuous discharge for 21 years, LK02 in the Liuzheng area will be the first well to be affected. With an increase in time, solutes will then gradually affect groundwater in some areas north of the Liuzheng water source.

### 4.4. Determination of Water Source Protection Areas

#### 4.4.1. Empirical Formula Method

The scope of the protection area of the Liuzheng water source is calculated using the empirical formula provided by the *Technical guideline for delineating source water protection areas* [44]:

$$R = \alpha \times K \times J \times T/n \tag{5}$$

where $R$ (m) is the radius of the protection zone, $\alpha$ is the safety factor, it is generally 150%; $K$ (m/day) is the aquifer permeability coefficient; $J$ is the hydraulic gradient; $T$ (day) is the horizontal migration time of the pollutant; and $n$ is effective porosity. According to the literature [44], when $T$ takes 100, $R$ is the radius of the primary protection area, and when $T$ takes 1000, $R$ is the radius of the secondary protection area.

According to the identification and verification of groundwater flow simulation; the permeability coefficient of the aquifer in the Liuzheng area is about 250 m/day. The average hydraulic gradient in the Liuzheng area is about 0.00035, the safety factor is 150%, and the effective porosity is 0.15. Using this information, results from Equation (5) show the radius of the primary protection zone to be 87.5 m, and the radius of the secondary protection zone to be 875 m.

#### 4.4.2. Numerical Simulation Method

Using the MODPATH module for backtracking, 10 tracer particles were set for each production well to predict the capture range of tracer particles passing through 100 and 1000 days (Figure 7). The maximum migration distance of the tracer particles for 100 day is about 90 m (Figure 7a), and the maximum migration distance of 1000 day outside the fault is about 760 m (Figure 7b).

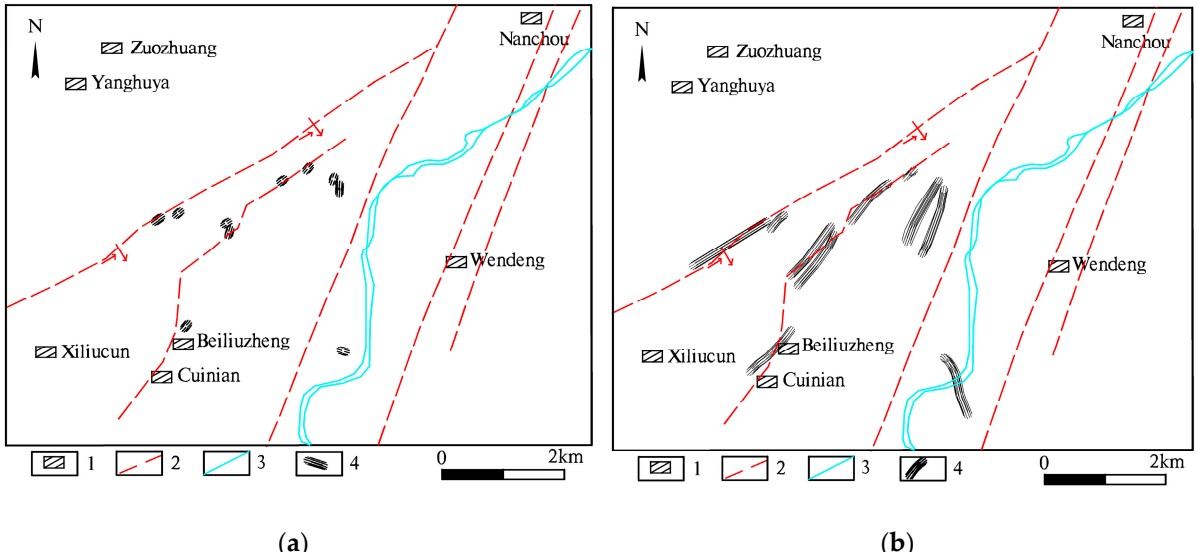

(**a**)　　　　　　　　　　　　　　　　　　　　　　　　(**b**)

**Figure 7.** Tracing trajectory of tracer particles. (**a**) 100 day, (**b**) 1000 day; 1: Village; 2: Fault; 3: River; 4: Migration trajectory.

#### 4.4.3. Comparative Analysis

For the division of the primary protection zone, the maximum migration distance obtained by the numerical simulation method is slightly larger than the radius calculated by the empirical formula method. For the division of the secondary protection zone, the maximum migration distance calculated by the numerical simulation method is slightly smaller than the radius calculated by the empirical formula method. The distance of the secondary protection zone calculated using the

numerical simulation method is slightly smaller than the radius of the protection zone calculated using the empirical formula method. This finding is due to the calculation of the empirical formula method, where hydraulic gradient takes the average hydraulic gradient in the Liuzheng area. In the numerical simulation method, the hydraulic gradient near the mining location is greater than the hydraulic gradient away from the mining well location. The numerical method not only simulates the actual situation of the groundwater flow field, it also shows the anisotropy of karst development in the partition of permeability coefficient, water storage coefficient and dispersion. For example, when the permeability coefficient near the fault zone is large, the radius of the protection zone is therefore also slightly large. Our results indicate that the numerical method is relatively accurate recording the area of the primary protection area of the Liuzheng water source to be about 0.59 km$^2$, and the area of the secondary protection area to be about 14.98 km$^2$ (Figure 8).

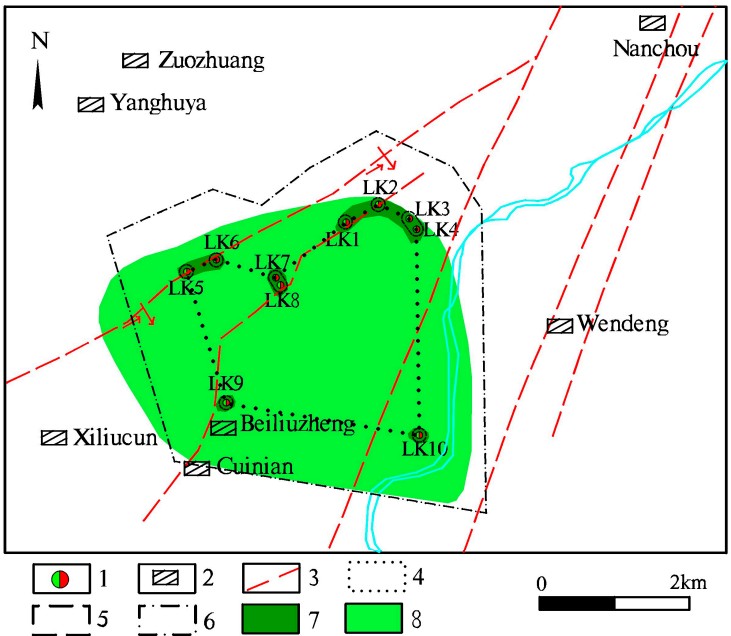

**Figure 8.** The scope of the protected area of the Liuzheng water source. 1: Mining wells; 2: Villages; 3: Faults; 4: Group well outsourcing line; 5: Primary protection area (experience formula method); 6: Secondary protection area (experience formula method); 7: Primary protection area (numerical simulation method); 8: Secondary protection area (numerical simulation method).

## 5. Conclusions

1. The tracer test results show that karst development in the Wangzhai and Hougao areas of Zibo are characterized by the coexistence of karst pores, karst fissures, fractures and pipelines, and that the degree of development is uneven. The runoff channel and the dispersion coefficient are anisotropic, and the overall dispersion coefficient is less than 15 m$^2$/day. In addition, the tracer concentration curve can be divided into three types: Single peak, double peak and triple peak. The Wangzhai experimental area is one of the recharge areas of the Dawu and Liuzheng water sources.

2. Using the numerical simulation optimization calculation, when extraction of the Liuzheng water source reaches 150,000 m$^3$/day, it will not only change the regional groundwater flow field in the Dawu karst water system, it will also make full use of groundwater resources. The solute transport model predicts that the proposed Wangzhai Chemical Industrial Park will have a certain potential pollution impact on the northern part of the Liuzheng water source area.

3. The empirical formula method and the numerical method can be used to divide the protected area of the Liuzheng water source, and to compare and analyze results of the protection area division

of the two methods. Our results indicated that the numerical simulation method is suitable for use in this area. The area of the primary protection area is recorded as being 0.59 km$^2$, and the area of the secondary protection area is 14.98 km$^2$. Corresponding anti-pollution protection measures should therefore be implemented for all levels of the protected areas in accordance with relevant national regulations.

**Author Contributions:** Conceptualization, H.Z.; Investigation, X.L. and N.B.; Methodology, Y.D. and L.X.; Resources, L.Y. and Z.L.; Writing—original draft, H.Z. and Y.D.; Writing—review & editing, L.X.

**Funding:** This article is supported by the National Natural Science Foundation of China (41772257) and the Doctoral Fund of Jinan University (XBS1817).

**Acknowledgments:** We acknowledge Guangyao Chi and Xinyu Hou of University of Jinan for their extensive input of ideas on this research. This project would not have been possible without the cooperation of the Shandong Institute of Geological Survey.

**Conflicts of Interest:** The authors declare no conflict of interest. The founding sponsors had no role in the design of the study; in the collection, analyses, or interpretation of data; in the writing of the manuscript, and in the decision to publish the results.

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
