# Peer review of "Protection of the Liuzheng Water Source: A Karst Water System in Dawu, Zibo, China"

_water, doi:10.3390/w11040698_

Round 1

Reviewer 1 Report

The manuscript "Protection of the Liuzheng Water Source: a Karst Water System in Dawu, Zibo, China"  is on a topic of interest to Water Journal.

The manuscript tackles an important issue regarding karst aquifers.

In order to take advantage of this potential, the authors have to put some more effort based on the following comments:

In the Introduction part: Discussing the karst aquifers characteristics need to be in more details (more references literature about karst aquifers in worldwide need to be added). For example: Hartmann et al., 2015; Maloszewske et al., 2002; Butscher and Huggenberger 2009.

In the Introduction part: More explanation and information about pollution problem in karst aquifers and pollutants travel time in karst aquifers need to be added to the introduction part. (For example: McGuire and McDonell 2006; Morales et al., 2007; Schilling and Wolter 2007; Hamdan et al., 2016).

There is no description for the study area regarding: Topography, Soil, Average rainfall, etc. Beside of that, more information regarding to the available karst features in the study area need to be mentioned.

There is no any figure showing the location of the study area in China. Figure 1 should be modified by adding China map in one of the upper corners and showing the location of the study area compared to China (Moving from regional to local scale).

Figure 1: The villages and source well symbols are not clear.

In the Method part: Time frequency of collecting samples after injecting the tracer need to be mentioned (hourly, daily, etc).

The wells WK01 and W512 mentioned in line 110 are very difficult to see in Fig 1. Both of the wells need to be more clearer in Fig 1.

In the Method part: What is the flow model cell size? please mention that in the text.

Figure 4: Regarding the water level lines, are they above sea level or depth to water (below ground level). Please mention that in the legend or in the figure caption.

Figure 5: villages symbol is not clear.

Figure 6: the units should be between brackets (time (d); CL- (mg.l-1)).

Figure 6 is not clear. It represents the increasing in chloride concentration in the water source mining wells. The author stopped the graph in the day around 1750. while in Fig 5, based on the plan range of solute migration of chloride, the wells will start affecting by chloride after 30, while Fig 5 covering just 1750 day (around 4.8 years). In addition. after 30 years, wells No. LK02, LK03, LK01, LK06 will start affecting by chloride, comparing with No 10 and 9 which is not affected even after 50 years. So why in Fig 6, the LK09 and LK10 wells curve are plotted while not for LK03, LK01 and LK06.

Figure 7: Symbols 1 and 4 are not clear.

References:

Hartmann A, Goldscheider N, Wagener T, Lange J, Weiler M (2014) Karst water resources in a changing world: Review of hydrological modeling approaches, Rev. Geophys., 52 : 218–242, doi:10.1002/2013RG000443.

Maloszewski P, Stichler W, Zuber A, Rank D (2002) Identifying the flow systems in a karstic-fissured-porous aquifer, the Schneealpe, Austria, by modelling of environmental 18O and 3H isotopes, Journal of Hydrology. 256(1-2): 48–59, doi:10.1016/S0022-1694(01)00526- 1.

Butscher C, Huggenberger P (2009) Modeling the temporal variability of karst groundwater vulnerability, with implication for climate change, Environmental Science and Technology. 43(6): 1665-1669, doi 10.1021/es801613g.

Hamdan I, Wiegand B, Toll M, Sauter M (2016a) Spring response to precipitation events using δ18O and δ2H in the Tanour catchment, NW Jordan, Isotopes in Environmental and Health Studies. 52(6): 682-693, doi: 10.1080/10256016.2016.1159205.

McGuire K, McDonnell J (2006) A review and evaluation of catchment transit time modeling, Journal of Hydrology. 330: 543-563, doi: 10.1016/j.jhydrol.2006.04.020.

Morales T, de Valderrama I, Uriarte J, Antiguedad I, Olazar M (2007) Predicting travel times and transport characterization in karst conduits by analyzing tracer-breakthrough curves, Journal of Hydrology. 334:183–198, doi:10.1016/j.jhydrol.2006.10.006.

Schilling KE, Wolter CF (2007) A GIS-based groundwater travel time model to evaluate stream nitrate concentration reductions from land use change, Environ Geol. 53: 433-443, doi 10.1007/s00254-007-0659-0.

Author Response

Dear editor/reviewer:

We thank the two reviewers and editor for their insightful and encouraging comments. We appreciate their efforts in helping to improve our paper. We have finished the correction according to the reviewers’ suggestions. Reviewers’ comments are in black font, our responses and any changes are in red font. All the corrections are clearly highlighted in red font in the manuscript as well. All the comments from the reviewers have been replied point by point. If you have any queries, please contact us.

Thank you and best regards.

Corresponding author: Liting Xing

Reviewer :

Point 1: In the Introduction part: Discussing the karst aquifers characteristics need to be in more details (more references literature about karst aquifers in worldwide need to be added). For example: Hartmann et al., 2015; Maloszewske et al., 2002; Butscher and Huggenberger 2009.

Response 1: According to your suggestion, and after extensive review of the literature, we have improved our discussion of the karst aquifers characteristics and pollution problem. We have added some sentences. “Karst landforms are widely distributed around the world, and karst water provides drinking water to nearly a quarter of the world's population. In China, the total area of bare limestone is about 1.3 million square kilometers, accounting for 13.5% of the total area of the country, and covered limestone buried underground is more extensive. During the long geological history evolution, landforms such as gully, dissolved depression, dry valley, blind valley, falling water cave, and skylight are formed. The soluble rocks can be dissolved and eroded by groundwater along the layers, joints or structural fractures, and formed underground passages, rapidly moving through karst fissures, karst conduits, etc. Groundwater in karst area is extremely easy to be polluted, which means that karst aquifer has a high degree of vulnerability [1-3]. The multiple conduit flow of karst medium leads to rapid contamination, while the diffuse flow forms persistent contamination [4]. Due to the uneven development of karst as well as the complex and variable movement of groundwater flow, people try to study the migration of pollutants in karst water from different scales using different methods, such as isotopes [5], tracer test [6], pumping test [7], numerical simulation [8-10], indoor experiment [11], and so on. For example, Hamdan et al. revealed the characteristics of groundwater vulnerability to pollution with the method of combining the stable isotopic (oxygen and hydrogen) and data of water temperature, spring discharge, and turbidity [12]. Morales et al. depicted transport characteristics along preferential flow paths in karst aquifers by analyzing data obtained in 26 tracer tests in Basque [13]. Maloszewski et al. combined application of lumped-parameter models to establish a measurement model for 18O and tritium in precipitation and springs, and obtained mean values of hydraulic parameters [14]. Therefore, in order to effectively manage vulnerable karst water systems, specific exploration techniques and modeling methods are required to master, studying the media characteristics and hydrodynamic field characteristics of the karst area [15-16].” (Line 37-59); “With the rapid development of computer science, numerical methods have been widely used. Numerical methods can not only simulate large-scale karst water movement and evaluate karst water resources [1], but also predict groundwater pollution processes [29].” (Line 71-74); Therefore, this paper uses tracer test to reveal the characteristics of medium field, uses numerical simulation to grasp the characteristics of groundwater flow field, and combines tracer test, empirical formula and numerical simulation to divide the protection area of karst water source area.” (Line 97-100).

Point 2: In the Introduction part: More explanation and information about pollution problem in karst aquifers and pollutants travel time in karst aquifers need to be added to the introduction part. (For example: McGuire and McDonell 2006; Morales et al., 2007; Schilling and Wolter 2007; Hamdan et al., 2016).

Response 2: According to your suggestion, and after extensive review of the literature, we have improved our discussion of the karst aquifers characteristics and pollution problem. We have added some sentences. “Karst landforms are widely distributed around the world, and karst water provides drinking water to nearly a quarter of the world's population. In China, the total area of bare limestone is about 1.3 million square kilometers, accounting for 13.5% of the total area of the country, and covered limestone buried underground is more extensive. During the long geological history evolution, landforms such as gully, dissolved depression, dry valley, blind valley, falling water cave, and skylight are formed. The soluble rocks can be dissolved and eroded by groundwater along the layers, joints or structural fractures, and formed underground passages, rapidly moving through karst fissures, karst conduits, etc. Groundwater in karst area is extremely easy to be polluted, which means that karst aquifer has a high degree of vulnerability [1-3]. The multiple conduit flow of karst medium leads to rapid contamination, while the diffuse flow forms persistent contamination [4]. Due to the uneven development of karst as well as the complex and variable movement of groundwater flow, people try to study the migration of pollutants in karst water from different scales using different methods, such as isotopes [5], tracer test [6], pumping test [7], numerical simulation [8-10], indoor experiment [11], and so on. For example, Hamdan et al. revealed the characteristics of groundwater vulnerability to pollution with the method of combining the stable isotopic (oxygen and hydrogen) and data of water temperature, spring discharge, and turbidity [12]. Morales et al. depicted transport characteristics along preferential flow paths in karst aquifers by analyzing data obtained in 26 tracer tests in Basque [13]. Maloszewski et al. combined application of lumped-parameter models to establish a measurement model for 18O and tritium in precipitation and springs, and obtained mean values of hydraulic parameters [14]. Therefore, in order to effectively manage vulnerable karst water systems, specific exploration techniques and modeling methods are required to master, studying the media characteristics and hydrodynamic field characteristics of the karst area [15-16].” (Line 37-59); “With the rapid development of computer science, numerical methods have been widely used. Numerical methods can not only simulate large-scale karst water movement and evaluate karst water resources [1], but also predict groundwater pollution processes [29].” (Line 71-74); Therefore, this paper uses tracer test to reveal the characteristics of medium field, uses numerical simulation to grasp the characteristics of groundwater flow field, and combines tracer test, empirical formula and numerical simulation to divide the protection area of karst water source area.” (Line 97-100).

Point 3: There is no description for the study area regarding: Topography, Soil, Average rainfall, etc. Beside of that, more information regarding to the available karst features in the study area need to be mentioned.

Response 3: According to your suggestion, the temperature, precipitation, topography and other information of the study area have been improved. “The study area belongs to warm temperate continental monsoon climate. The annual average temperature is 12.3 - 13.1 °C, and the average annual precipitation is 640.50 mm. The rivers in the area are all rain-source type, and the terrain is high in the south and low in the north. The terrain in the south is low mountain and hilly, and the highest elevation is 380 m. The northern part is connected to the sloped plain in front of mountain, and the lowest elevation is 40 m.” (Line 102-106)

Point 4: There is no any figure showing the location of the study area in China. Figure 1 should be modified by adding China map in one of the upper corners and showing the location of the study area compared to China (Moving from regional to local scale).

Response 4: Figure 1 has been modified.

Point 5: Figure 1: The villages and source well symbols are not clear.

Response 5: Figure 1 has been modified.

Point 6: In the Method part: Time frequency of collecting samples after injecting the tracer need to be mentioned (hourly, daily, etc).

Response 6: According to your suggestion, time frequency of collecting samples after injecting the tracer has been mentioned. “After the tracer was injected, in the encrypted observation area within 2 km from the source point, the monitoring frequency was once a day, the area outside the encryption area was sampled once every three days. Monitoring holes close to the source well were sampled first. As the tracer migrated, the sampling frequency of the surrounding area was gradually encrypted.” (Line 144-148)

Point 7: The wells WK01 and W512 mentioned in line 110 are very difficult to see in Fig 1. Both of the wells need to be more clearer in Fig 1.

Response 7: Figure 1 has been modified.

Point 8: In the Method part: What is the flow model cell size? please mention that in the text.

Response 8: According to your suggestion, we have added a description of the flow model cell size cell. “The planar area was divided by a discrete method of rectangular finite difference and the cell size was 500 m × 500 m.” (Line 162-163)

Point 9: Figure 4: Regarding the water level lines, are they above sea level or depth to water (below ground level). Please mention that in the legend or in the figure caption.

Response 9: The water level lines are above sea level. Legend 3 has been changed to “groundwater level contour (m)”. (Line 263)

Point 10: Figure 5: villages symbol is not clear.

Response 10: Figure 5 has been modified.

Point 11: Figure 6: the units should be between brackets (time (d); Cl- (mg.l-1)).

Response 11: Figure 6 has been modified.

Point 12: Figure 6 is not clear. It represents the increasing in chloride concentration in the water source mining wells. The author stopped the graph in the day around 1750. while in Fig 5, based on the plan range of solute migration of chloride, the wells will start affecting by chloride after 30, while Fig 5 covering just 1750 day (around 4.8 years). In addition. after 30 years, wells No. LK02, LK03, LK01, LK06 will start affecting by chloride, comparing with No 10 and 9 which is not affected even after 50 years. So why in Fig 6, the LK09 and LK10 wells curve are plotted while not for LK03, LK01 and LK06.

Response 12: (1) The time in Figure 6 is not stopped at 1750. Figure 6 predicted the concentration of Cl- during 18 263 days (50 years). (2) Figure 6 has supplemented the predicted concentration curve of Cl- in Lk1, Lk3, Lk4, Lk6 and Lk8 wells.

Point 13: Figure 7: Symbols 1 and 4 are not clear.

Response 13: Figure 7 has been modified.

In addition, (1) Figures 2, 3, 4, and 8 have also been appropriately modified. (2) “2. Geological background of the study area” has been changed to “2. Study area”.

Reviewer 2 Report

This is an interesting paper, but it needs to be improved:

·         text and equations need further editing: spacing, inconsistent italic notation, the mixup of upper and lower cases

·         some parts of the text should be explained in more details

·         the titles of certain tables do not match the contents of the table

·         the reference list does not correspond to the requirements of the journal

Detailed comments are in the enclosed pdf.

Author Response

Dear editor/reviewer:

We thank the two reviewers and editor for their insightful and encouraging comments. We appreciate their efforts in helping to improve our paper. We have finished the correction according to the reviewers’ suggestions. Reviewers’ comments are in black font; our responses and any changes are in red font. All the corrections are clearly highlighted in red font in the manuscript as well. All the comments from the reviewers have been replied point by point. If you have any queries, please contact us.

Thank you and best regards.

Corresponding author: Liting Xing

Reviewer :

Point 1:Andreo” “Dmitry”

Response 1: “Andreo” has been changed to “Andreo et al.” (Line 70), “Dmitry” has been changed to “Rudakov et al.” (Line 74). In addition, “Rao Lei” has been changed to “Rao Lei et al.” (Line 77)

Point 2: “high in the south and low in the north”——indicate approximate elevations

Response 2: According to your suggestion, a sentence about elevations has been added. “The terrain in the south is low mountain and hilly, and the highest elevation is 380 m. The northern part is connected to the sloped plain in front of mountain, and the lowest elevation is 40 m.” (Line 104-106)

Point 3: “Quaternary”

Response 3: “Quaternary” has been changed to “Quaternary loose rock strata”. (Line 109)

Point 4: “the principle of near-far and long-distance was adopted——expalin or add citation

Response 4: According to your suggestion, the sentence “After the tracer was injected, in the encrypted observation area within 2 km from the source point, the monitoring frequency was once a day, the area outside the encryption area was sampled once every three days. Monitoring holes close to the source well were sampled first. As the tracer migrated, the sampling frequency of the surrounding area was gradually encrypted.” (Line 144-148)

Point 5:Formula (1)——inconsistent italic notation

Response 5: Formula (1) has been modified. At the same time, the following “μ” has been changed to “μ”. (Line 157)

Point 6:Formula (2)——inconsistent italic notation inconsistent uppercase and lowercase, Qs is not part of the equation (2)

Response 6: Formula (2) has been modified. At the same time, the following “c” has been changed to“C”, “Qs” has been changed to“qs”, “ΣRk” has been changed to“ΣRk”. (Line 173-175)

Point 7: “Table 1. Karst development characteristics in the test area” ——The title does not match the content in the table

Response 7: The sentence “Karst development characteristics in the test area” has been changed to “Tracer concentration curve type of observation well.” (Line 194)

Point 8: “The various types of curves reflect the difference in the degree of aquifer fissure development, with a single channel and two or three main channels.” ——Strong assertion. Explain in detail or quote the appropriate reference.

Response 8: This position has added a new reference. “[41]” (Line 201), “Hengxiang, Z.; Liting, X.; Hua, X.; Guangyao, C.; Xinyu, H. Application of tracer test in the study of preferential runoff path of Jinan spring group. Ground water 2017, 39, 5-7.” (Line 454-455)

Point 9: “(Fig. 4).” ——Describe this figure in the text.

Response 9: According to your suggestion, the sentence “and the natural groundwater flow in the regional groundwater will not change significantly (Fig. 4).” has been changed to “Figure 4 showed the groundwater flow field of Liuzheng area during wet season after adjusting the mining layout. There is no significant change in regional natural groundwater flow field. The average water level in the mining center dropped from about 27 m adjusting the mining layout to about 22 m after adjusting the mining layout, that is, the mean water level drop was about 5  m, and there will not appear hydrogeological problems such as karst collapse.” (Line 254-259)

Point 10: “LK11 well” ——In the text, explain why you took this well as an example.

Response 10: The LK11 well is located at the center of the funnel and is sensitive to the reaction of artificial mining. It can predict the maximum impact of mining on groundwater level. Therefore, the groundwater dynamics are predicted by using LK11 well as an example.

Point 11: “Technical guideline for delineating source water protection areas (HJ338-2018):” —— Wrong quotation

Response 11: The quotation has been modified. “Technical guideline for delineating source water protection areas [44]: ” (Line 288)

Point 12: “When the primary protection area is divided” ——Divided how?

Response 12: The sentence has been changed to “According to the literature [44], when T takes 100, R is the radius of the primary protection area, and when T takes 1000, R is the radius of the secondary protection area.” (Line 292-294)

Point 13: “Division results for the empirical formula method and the numerical simulation method record a slightly larger maximum distance for the primary protection zone using the numerical simulation method than the radius of the protection zone calculated using the empirical formula method.” ——It isn't clear what you mean. Explain clearer.

Response 13: The sentence has been changed to “For the division of the primary protection zone, the maximum migration distance obtained by numerical simulation method is slightly larger than the radius calculated by empirical formula method. For the division of the secondary protection zone, the maximum migration distance calculated by numerical simulation method is slightly smaller than the radius calculated by empirical formula method.” (Line 310-314)

Point 14:References——Thesis and reports are not listed properly

Response 14: The references have been modified. (1) “Yang, Z. Shallow groundwater quality and evaluation in karst areas of urbanization. Master thesis, Guizhou University, Guiyang, 2009.” (Line 398-399); “Mei, Y. Controlling factor and migration characteristic research of organic pollutant in typical karst underground rivers: A case of Nanshan karst valley in Chongqing. Master thesis, Southwest University, Chongqing, 2010.” (Line 403-405); “Hao, C. Study on delineating groundwater source protection zones in Sihe alluvial fan. Master thesis, China University of Geosciences, Beijing, 2018.” (Line 438-439); “Feng, X. Study of groundwater contamination transport simulation on a project in southwest karst area based on Visual MODFLOW. Master thesis, Hefei University of Technology, Hefei, 2014.” (Line 451-452). (2) “Shandong Institute of Geological Survey. Hydrogeological exploration report of Liuzheng water source. Shandong Institute of Geological Survey, Jinan, 2018.” (Line 449-450); “Shandong Institute of Geological Survey. Prediction of the impact of the new site planning of the chemical industry enterprises in Zhangdian on the groundwater environment. Shandong Institute of Geological Survey, Jinan, 2011.” (Line 453-455). (3) Rudakov, D.V.; Rudakov, V.C. Analytical modeling of aquifer pollution caused by solid waste depositories. Ground Water 1999, 37, 352-357. (Line 376-377).

In addition, after a lot of reading for the literature, in the introduction we have added some literature related to the content of this article. For details, please see the line 37-59 of the revised manuscript.

Round 2

Reviewer 1 Report

The manuscript has been significantly improved, and the author's took all the comments into consideration.